# AN INTERPRETABLE LSTM NEURAL NETWORK FOR AUTOREGRESSIVE EXOGENOUS MODEL

**Tian Guo** [*]
ETH Zurich, Switzerland
{tian.guo}@gess.ethz.ch

**Tao Lin** [*]
EPFL Lausanne, Switzerland
{tao.lin}@epfl.ch

**Yao Lu**
Tendcloud Beijing, China
{yao.lu}@tendcloud.com

## ABSTRACT

In this paper, we propose an interpretable LSTM recurrent neural network, i.e., multi-variable LSTM for time series with exogenous variables. Currently, widely used attention mechanism in recurrent neural networks mostly focuses on the temporal aspect of data and falls short of characterizing variable importance. To this end, our multi-variable LSTM equipped with tensorized hidden states is developed to learn variable specific representations, which give rise to both temporal and variable level attention. Preliminary experiments demonstrate comparable prediction performance of multi-variable LSTM w.r.t. encoder-decoder based baselines. More interestingly, variable importance in real datasets characterized by the variable attention is highly in line with that determined by statistical Granger causality test, which exhibits the prospect of multi-variable LSTM as a simple and uniform end-to-end framework for both forecasting and knowledge discovery.

## 1 INTRODUCTION

Time series, a sequence of observations over time, is being generated in a wide variety of areas (Qin et al., 2017; Lin et al., 2017a). Long short-term memory units (LSTM) (Hochreiter & Schmidhuber, 1997) and the gated recurrent unit (GRU) (Cho et al., 2014) have achieved great success in various applications on sequence data because of the gate and memory mechanism (Wang et al., 2016; Lipton et al., 2015).

In this paper, we focus on time series with exogenous variables. Specifically, given a target time series, we have an additional set of time series corresponding to exogenous variables. A predictive model using the historical data of both target and exogenous variables to predict the future values of the target variable is an autoregressive exogenous model, referred to as ARX. In addition to forecasting, it is also highly desirable to distill knowledge via the model, e.g., understanding the different importance of exogenous variables w.r.t. the evolution of the target series (Hu et al., 2018; Siggiridou & Kugiumtzis, 2016; Zhou et al., 2015). However, current LSTM RNN falls short of such capability. When fed with the historical observations of the target and exogenous variables, LSTM blindly blends the information of all variables into the hidden states and memory cells for subsequent prediction. Therefore, it is intractable to distinguish the contribution of individual variables by looking into hidden states (Zhang et al., 2017).

Recently, attention-based neural networks (Bahdanau et al., 2014; Xu et al., 2015; Chorowski et al., 2015) have been proposed to enhance the ability of LSTM in using long-term memory as well as the interpretability. The attention mechanism is mostly applied to hidden states across time steps, thereby solely uncovering the temporal level importance rather than the variable level importance.

To this end, we propose an interpretable LSTM recurrent neural network, called multi-variable LSTM, for ARX problem. A distinguishing feature of our multi-variable LSTM is to enable each

---

[*]Equal contribution.

neuron of the recurrent layer to encode information exclusively from a certain variable. As a result, from the overall hidden states of the recurrent layer, we derive variable specific hidden representations over time steps, which can be flexibly used for forecasting and temporal-variable level attentions.

## 2 RELATED WORK

The success of attention mechanism proposed in (Bahdanau et al., 2014) has motivated a wide use of attention in image processing (Ba et al., 2014; Mnih et al., 2014; Gregor et al., 2015; Xu et al., 2015), natural language processing (Hermann et al., 2015; Rush et al., 2015; Lin et al., 2017b) and speech recognition (Chorowski et al., 2015). However, traditional attention mechanism is normally applied to hidden states across time steps, thereby failing to reveal variable level attention. Only some very recent studies (Choi et al., 2016; Qin et al., 2017) attempted to develop attention mechanism to handle multi-variable sequence data. Both of them build on top of encoder-decoder architecture, and make the prediction using pre-weighted input obtained from the encoder or an additional RNN. Our MV-LSTM is a simple one recurrent layer architecture enabling variable specific representations.

## 3 MULTI-VARIABLE LSTM

In this section, we present the proposed multi-variable LSTM referred to as MV-LSTM in detail.

Assume we have $N-1$ exogenous time series and target series $\mathbf{y}$ of length $T$, where $\mathbf{y} = (y_1, \cdots, y_T)$ and $\mathbf{y} \in \mathbb{R}^T$.[1] By stacking exogenous time series and target series, we define the multi-variable input of MV-LSTM at each time step as $\mathbf{X} = (\mathbf{x}_1, \ldots, \mathbf{x}_T)$, where $\mathbf{x}_t = (x_{t,1}, \ldots, x_{t,N-1}, y_t) \in \mathbb{R}^N$ and $x_{t,n} \in \mathbb{R}$ is the observation of $n$-th exogenous time series at time $t$. Given $\mathbf{X}$, we aim to learn a non-linear mapping to the one-step ahead value of the target series, namely $\hat{y}_{T+1} = \mathcal{F}(\mathbf{X})$, where $\mathcal{F}(\cdot)$ represents the MV-LSTM neural network we present below.

Inspired by (He et al., 2017), our MV-LSTM has tensorized hidden states and the update scheme ensures that each element of the hidden state tensor encapsulates information exclusively from a certain variable of the input.

Specifically, we define the hidden state and memory cell for $t$-th time step of MV-LSTM as $\mathbf{h}_t \in \mathbb{R}^M$ and $\mathbf{c}_t \in \mathbb{R}^M$, where $M$ is the size of recurrent layer. $\mathbf{h}_t$ is tensorized as $\mathcal{H}_t = [\mathbf{h}_t^1, \ldots, \mathbf{h}_t^N]^\top$, where $\mathcal{H}_t \in \mathbb{R}^{N \times d}$, $\mathbf{h}_t^n \in \mathbb{R}^d$ and $N \cdot d = M$. The element $\mathbf{h}_t^n$ of tensor $\mathcal{H}_t$ is a variable specific representation corresponding to $n$-th input dimension. We further define the input-to-hidden transition tensor as $\mathcal{W}_x = [\mathbf{W}_x^1, \ldots, \mathbf{W}_x^N]^\top$, where $\mathcal{W}_x \in \mathbb{R}^{N \times d}$ and $\mathbf{W}_x^n \in \mathbb{R}^d$. The hidden-to-hidden transition tensor is defined as: $\mathcal{W}_h = [\mathbf{W}_h^1, \ldots, \mathbf{W}_h^N]^\top$, where $\mathcal{W}_h \in \mathbb{R}^{N \times d \times d}$ and $\mathbf{W}_h^n \in \mathbb{R}^{d \times d}$.

Given the new incoming input $\mathbf{x}_t$ and the hidden state $\mathbf{h}_{t-1}$ up to $t-1$, we formulate the iterative update process by using $\mathcal{W}_x$ and $\mathcal{W}_h$ as:

$$\mathbf{j}_t = \tanh(\mathcal{H}_{t-1} * \mathcal{W}_h + \mathbf{x}_t * \mathcal{W}_x + \mathbf{b}_j) = \tanh \left( \begin{bmatrix} (\mathbf{W}_h^1 \mathbf{h}_{t-1}^1)^\top \\ \vdots \\ (\mathbf{W}_h^N \mathbf{h}_{t-1}^N)^\top \end{bmatrix} + \begin{bmatrix} (\mathbf{W}_x^1 x_{t,1})^\top \\ \vdots \\ (\mathbf{W}_x^N x_{t,N})^\top \end{bmatrix} + \mathbf{b}_j \right)$$

where $*$ represents the element-wise multiplication operation on tensor elements. $\mathcal{H}_{t-1} * \mathcal{W}_h \in \mathbb{R}^{N \times d}$ is the concatenation of $N$ product results of hidden tensor element $\mathbf{h}_t^n$ and the corresponding transition matrix $\mathbf{W}_h^n$. Likewise, $\mathbf{x}_t * \mathcal{W}_x \in \mathbb{R}^{N \times d}$, and represents how the current input $\mathbf{x}_t$ update the hidden state.

$\mathbf{j}_t$ is an $N \times d$ dimensional tensor, and each $d$-dimensional element corresponds to one input variable, encoding the information exclusively from the variable specific hidden state and the input variable.

The input, forget and output gates in MV-LSTM are updated by using all input dimensions of $\mathbf{x}_t$, so as to utilize the cross-correlation between multi-variable time series. In particular, $[\mathbf{i}_t, \mathbf{f}_t, \mathbf{o}_t]^\top = \sigma\left(\mathbf{W}[\mathbf{x}_t, \mathbf{h}_{t-1}] + \mathbf{b}\right)$. The updated memory cell and hidden states are obtained from $\mathbf{c}_t = \mathbf{f}_t \odot \mathbf{c}_t + \mathbf{i}_t \odot \tilde{\mathbf{j}}_t$, where $\tilde{\mathbf{j}}_t \in \mathbb{R}^M$ is the flattened vector of $\mathbf{j}_t$. Then, $\mathbf{h}_t = \mathbf{o}_t \odot \tanh(\mathbf{c}_t)$.

---

[1] Vectors are assumed to be in column form throughout this paper.

After feeding $\mathbf{x}_T$ into MV-LSTM, we obtain the hidden representation $\mathbf{h}_T^n$ w.r.t. each variable, which can be combined with the attention mechanism to predict $y_{T+1}$ as well as interpreting variable importance. Concretely, the attention process is as: $e^n = \tanh(\mathbf{W}_e \mathbf{h}_T^n + b_e)$ and $\alpha^n = \frac{exp(e^n)}{\sum_{k=1}^{N} exp(e^k)}$. Then, the prediction is derived as: $\hat{y}_{T+1} = \sum_{n=1}^{N} \alpha^n (\mathbf{W_n} \mathbf{h}_n^T + b_n)$. Note that MV-LSTM is able to apply temporal attention with ease. In the present work, we focus on evaluating variable attention.

## 4 EXPERIMENTS

In this part, we report some preliminary results to demonstrate the prediction performance of MV-LSTM as well as the variable importance interpretation. Please refer to the appendix section for more results about MAE errors and variable attention interpretation.

We use two real datasets. **PM2.5:** It contains hourly PM2.5 data and the associated meteorological data in Beijing of China. PM2.5 measurement is the target series. The exogenous time series include dew point, temperature, pressure, combined wind direction, cumulated wind speed, hours of snow, and hours of rain. **ENERGY:** It collects the appliance energy use in a low energy building. The target series is the energy data logged every 10 minutes. Exogenous time series consist of 14 variables, e.g., house inside temperature conditions and outside weather information including temperature, wind speed, humanity and dew point from the nearest weather station.

Baselines include ensemble methods on time series, i.e., random forests (RF) (Liaw et al., 2002; Meek et al., 2002) and extreme gradient boosting (XGT) (Chen & Guestrin, 2016; Friedman, 2001), and state-of-the-art attention based recurrent neural networks on multi-variable sequence data, referred to as DUAL (Qin et al., 2017) and RETAIN (Choi et al., 2016).

In the first group of experiments, we report the prediction performance of all approaches in Table 1.

Table 1: Test errors (RMSE)

| Dataset | RF | XGT | DUAL | RETAIN | MV-LSTM |
|---------|-----|-----|------|--------|---------|
| PM2.5 | $0.573 \pm 0.003$ | $0.370 \pm 0.002$ | $0.355 \pm 0.002$ | $1.112 \pm 0.017$ | $0.340 \pm 0.001$ |
| ENERGY | $0.494 \pm 0.004$ | $0.360 \pm 0.003$ | $0.372 \pm 0.006$ | $0.669 \pm 0.006$ | $0.361 \pm 0.001$ |

Next, we analyze the variable level attention obtained in MV-LSTM on PM2.5 dataset. Specifically, in testing phase MV-LSTM outputs variable attention values specific for each testing instance, and thus for each variable we can estimate an empirical distribution of the corresponding attention value. Figure 1 shows the histograms of the attention values on top four variables, which are ranked by the empirical mean of their attention values. For comparison, variables identified by Granger causality test (Arnold et al., 2007) w.r.t. the target variable are shown with a colored background.

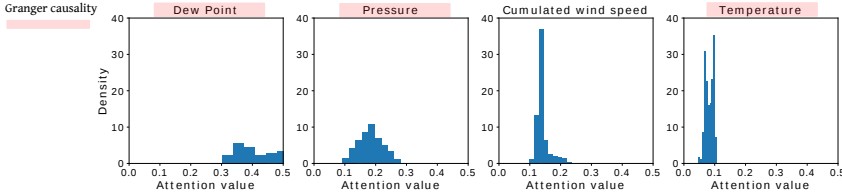

Figure 1: Top four variables ranked by empirical mean of attention values in MV-LSTM on PM2.5 dataset. Variable names with colored background indicates the identification by Granger-causality as well.

We observe in Figure 1 that three variables (i.e., Dew point, pressure, and temperature) identified by Granger causality test are also top ranked by the variable attention of MV-LSTM. As pointed out by (Liang et al., 2015), dew point and pressure are usually affected by the arrival of the northerly wind, which brings in drier and fresher air. This is exactly in line with the observation in the rank of such three variables by MV-LSTM. On the contrary, DUAL fails to reveal meaningful variable importance, as is shown in the appendix section.

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

# 5 APPENDIX

Table 2: Test errors (MAE)

| Dataset | RF | XGT | DUAL | RETAIN | MV-LSTM |
|---------|-----|------|------|--------|---------|
| PM2.5 | $0.433 \pm 0.011$ | $0.302 \pm 0.012$ | $0.248 \pm 0.003$ | $0.943 \pm 0.018$ | $0.227 \pm 0.002$ |
| ENERGY | $0.404 \pm 0.021$ | $0.310 \pm 0.023$ | $0.249 \pm 0.006$ | $0.507 \pm 0.016$ | $0.256 \pm 0.006$ |

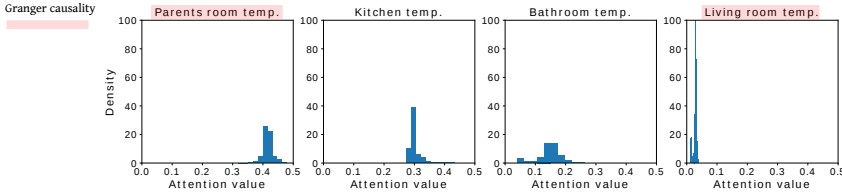

Figure 2: Top four variables ranked by empirical mean of attention values in MV-LSTM on EN-ERGY dataset. Variable name with the color background indicate that the variable is identified by Granger-causality test as well. Only variable parents room temp. and living room temp. are identified by Granger-causality test in this dataset.

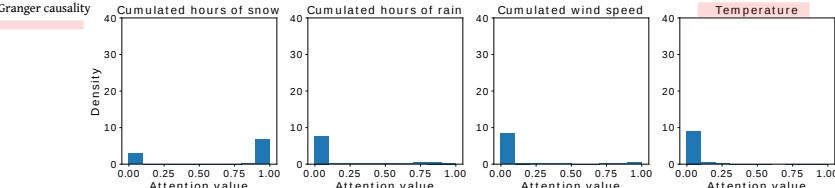

Figure 3: Top four variables ranked by empirical mean of attention values in DUAL on PM2.5 dataset. Variable name with the color background indicate that the variable is identified by Granger-causality test as well.

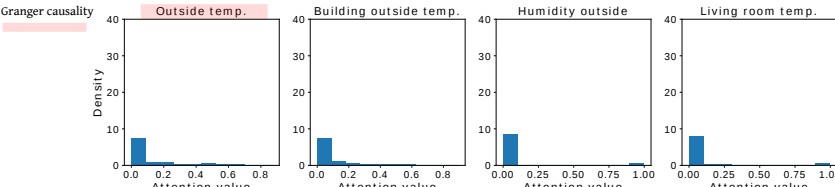

Figure 4: Top four variables ranked by empirical mean of attention values in DUAL on ENERGY dataset. Variable name with the color background indicate that the variable is identified by Granger-causality test as well.

