# OpenReview forum: "An interpretable LSTM neural network for autoregressive exogenous model"
_ICLR.cc/2018/Workshop — Accept_

### Official Review · AnonReviewer2 · 2018-03-10
**Better model for exogenous time series**

**Rating:** 7
**Confidence:** 4

**Review:**

The paper is focused on sequential data with exogenous time series. Previous attention-based models on this kind of dataset used multiple RNNs for each time series (Choi et al., 2016; Qin et al., 2017). The paper instead proposes to use single RNN to model the data by decomposing the hidden state of the RNN into several sub-states, each of which corresponding to each time series, which are individually attended but concatenated for the recurrent update. The paper shows statistically significant error reduction compared to the baselines on two datasets. The visualization also seems to be convincing. The validity of the paper's hypothesis on the model could be stronger if the model is tested on benchmarked datasets (it seems the datasets used in the paper are not used by previous papers?).

---

### Official Review · AnonReviewer3 · 2018-03-10
**Multivariate LSTM for time-series prediction**

**Rating:** 7
**Confidence:** 4

**Review:**

The paper studies the problem of time-series prediction given auxiliary variables. Instead of joining variables together as in standard RNNs, the paper allocates states and parameters per variable, and at the same time allows for cross-correlation among variables. This makes attention to variable importance easier to model. Experiments demonstrate the effectiveness of the proposed approach.

Overall the paper is presented relatively well, and there are elements of novelty in the model architecture in the context of time-series prediction. A figure of the multivariate LSTM may assist readers a bit more. A related work is matrix-centric RNN, studied in: https://arxiv.org/abs/1703.01454

---

### Official Review · AnonReviewer1 · 2018-03-11
**interesting idea, but hard to follow in section 3**

**Rating:** 4
**Confidence:** 4

**Review:**

Although I am not familiar with the ARX problem, but I agree on the limitation of LSTM mentioned in this paper. To my understanding, the idea of this paper is interesting.

On the other hand, I find it is really hard to follow the technical detail in section 3, especially the explanation of the equation of computing j_t. Without a good understanding of the technical part, I am not sure I can appreciate the value of the modeling part.

---

### Decision · Program_Chairs · 2018-03-20
**ICLR 2018 Workshop Acceptance Decision**

**Decision:**

Accept

**Comment:**

Congratulations, your paper was accepted to the ICLR workshop.